# Machine learning for buildings' characterization and power-law recovery of urban metrics

**Alaa Krayem[1], Aram Yeretzian[2], Ghaleb Faour[3], Sara Najem[1]***

**1** Physics Department, American University of Beirut, Beirut, Lebanon, **2** Architecture and Design, American University of Beirut, Beirut, Lebanon, **3** National Center for Remote Sensing, CNRS-L, Beirut, Lebanon

* sn62@aub.edu.lb

## Abstract

In this paper we focus on a critical component of the city: its building stock, which holds much of its socio-economic activities. In our case, the lack of a comprehensive database about their features and its limitation to a surveyed subset lead us to adopt data-driven techniques to extend our knowledge to the near-city-scale. Neural networks and random forests are applied to identify the buildings' number of floors and construction periods' dependencies on a set of shape features: area, perimeter, and height along with the annual electricity consumption, relying a surveyed data in the city of Beirut. The predicted results are then compared with established scaling laws of urban forms, which constitutes a further consistency check and validation of our workflow.

**Data Availability Statement:** All relevant data are available from https://zenodo.org/record/4001720#.X0ZaeZMzblw.

**Funding:** The author(s) received no specific funding for this work.

## Introduction

A key question for planning, designing, and managing urban spaces is how the different city's components interact and influence its dynamics [1]. It is therefore important to recognize cities as complex systems with emergent properties resulting form far from equilibrium dynamics and with energy requirements for self-maintenance [1, 2].

Specifically, urban form's evolution, or equivalently the city's infrastructure's spatial patterns, is governed by the rules of competitive processes which manifest themselves in self-similar fractal patterns or scaling laws, which govern the changes of city components with its size [1, 3–7]. At the micro-scale, urban buildings, which are described as "containers of socio-economic activities" are of particular interest [8]. Comprehensive surveys of buildings detailing their uses, ages, and sizes are essential to support more effective policymaking relating to the sustainable management of cities [9]. For instance, much of the work on the building stock has been driven by the energy sector. Particularly, urban buildings account for a high portion of the Green House Gas emissions through electricity consumption [10, 11]. Therefore, identifying building attributes helps in simulating their energy performance, identifying spatiotemporal patterns to assess the impact of retrofitting strategies to reduce energy consumption, and in adapting buildings for climate change [12–16]. Moreover, tracking the rate of change of cities and the survival of buildings is essential to estimate the distribution and lifecycle of stock

**Competing interests:** The authors have declared that no competing interests exist.

material in order to inform on the best practices of its management and utilization or the so-called "industrial ecology" [17, 18].

Therefore, generating a building database is important for urban science broadly speaking. Methodologically, it can rely on collecting existing information as in the property taxation database [19] or conducting ground surveys. However, this data is sometimes expensive, unavailable or insufficient. For this reason, taking advantage of new data sources, methods and tools is a central focus area in urban research. Volunteered Geographic Information (VGI) platforms, which are crowdsourcing tools, are gaining emerging interest. Coloring London [20], and Coloring Beirut [21], where residents are encouraged to fill, substitute, and update the buildings database themselves are such examples. Moreover, the automated capture and extraction of building attribute data are more and more facilitated by the development of computational resources, machine learning techniques, and remote sensing [9, 22–24].

In this paper, we apply machine learning algorithms to assist and complement the collection of building data. Neural Networks and Random Forests are built to link the physical character of the buildings to their number of floors and vintage respectively. We first outline the database we are working on, and then proceed by developing the machine learning algorithms for the buildings' attributes prediction. The relation of the results to established urban scaling is illustrated. Finally, we stress the importance of such methodology in data-scare environments in promoting transparency despite the challenges and bureaucratic impediments, which stand in the way of forming a national repository.

## Data collection and preprocessing

Beirut, the city, is located on the eastern shore of the Mediterranean sea with a stock of 17, 742 buildings (in 2016). The latter's corresponding footprints' attributes used in this work (area, perimeter and height), were obtained from the National Center for Scientific Research CNRS Lebanon, while additional information on a subset of 7, 122 buildings (among the 17, 742 buildings) was surveyed by the Saint-Joseph University (USJ), part of the LIBRIS program [ANR- LIBRIS project (ANR-09-RISK-006)—contribution to seismic hazard assessment in Lebanon. A co-joint project between ISTERRE, IPGP, EDYTEM and RESONNANCE laboratories with the AUB, NDU, USJ universities and the CNRS-L. Collaboration under the task 1.3. "Speleoseismicity and the Lebanese endokarst"]. It includes buildings' year of construction, type, number of floors and of apartments. Their corresponding construction years were converted into construction periods based on the city's architectural history, which witnessed five major waves of construction each with specific distinctive features [16, 25, 26]. The distribution of the USJ subset according to the year of construction is given in Table 1.

Moreover, data on the annual electricity consumption for many buildings were obtained from the national power utility: Electricité du Liban (EDL). Entries with missing fields, incorrect buildings' heights ($\leq 2.8m$), or atypical floor height ($\leq 2.8m$ or $\geq 4, 5m$) were removed from the dataset. It is worth noting that buildings' footprints were manually digitized over the

**Table 1. Percentage of buildings per period of construction in the dataset.**

| Construction period | Label | Percentage of buildings |
|:---:|:---:|:---:|
| Before 1923 | 1 | 1.2% |
| 1924-1940 | 2 | 7.8% |
| 1941-1960 | 3 | 42.1% |
| 1961-1990 | 4 | 39.1% |
| After 1991 | 5 | 9.7% |

entire city of Beirut by CNRS-L using aerial photos at 15$cm$ resolution and VHR pan-chromatic satellite images from Pleaides-1A at 70$cm$ resolution. This is not part of our work but rather a CNRS-L in-house processing step, which was made available to us. The floor height ranges were also determined as in [25], where a height of 4.5$m$ corresponds to a floor of a building constructed before 1923, while a height of 2.8$m$ mainly corresponds to recent buildings. Residential and mixed buildings types were kept while others such as hospitals, places of worship, and schools were removed, which left us with 1, 968 buildings of the USJ dataset.

This step was followed by the application of an Isolation forests (iForest) scheme [27], which is an outliers' detection procedure and is essential to the removal of noisy, incorrect or aberrant information in the dataset. The outliers' removal was based on a set of features: the yearly electricity consumption, floors' number, height, perimeter, area, period of construction, and type of the building. Subsequently, 434 buildings were classified as outliers and were therefore removed from the dataset, leaving us with a dataset of 1, 534 buildings, which we used in what follows. To visualize the outliers, the six features were reduced to three using the Principle Component Analysis (PCA). The correlations between each new dimension and the two others were then illustrated using a two-dimensional scatter plot matrix shown in Fig 1. The diagonal plots show the univariate distribution of each dimension. The spatial distribution of the buildings used in the development of the predictive algorithms is shown in Fig 2.

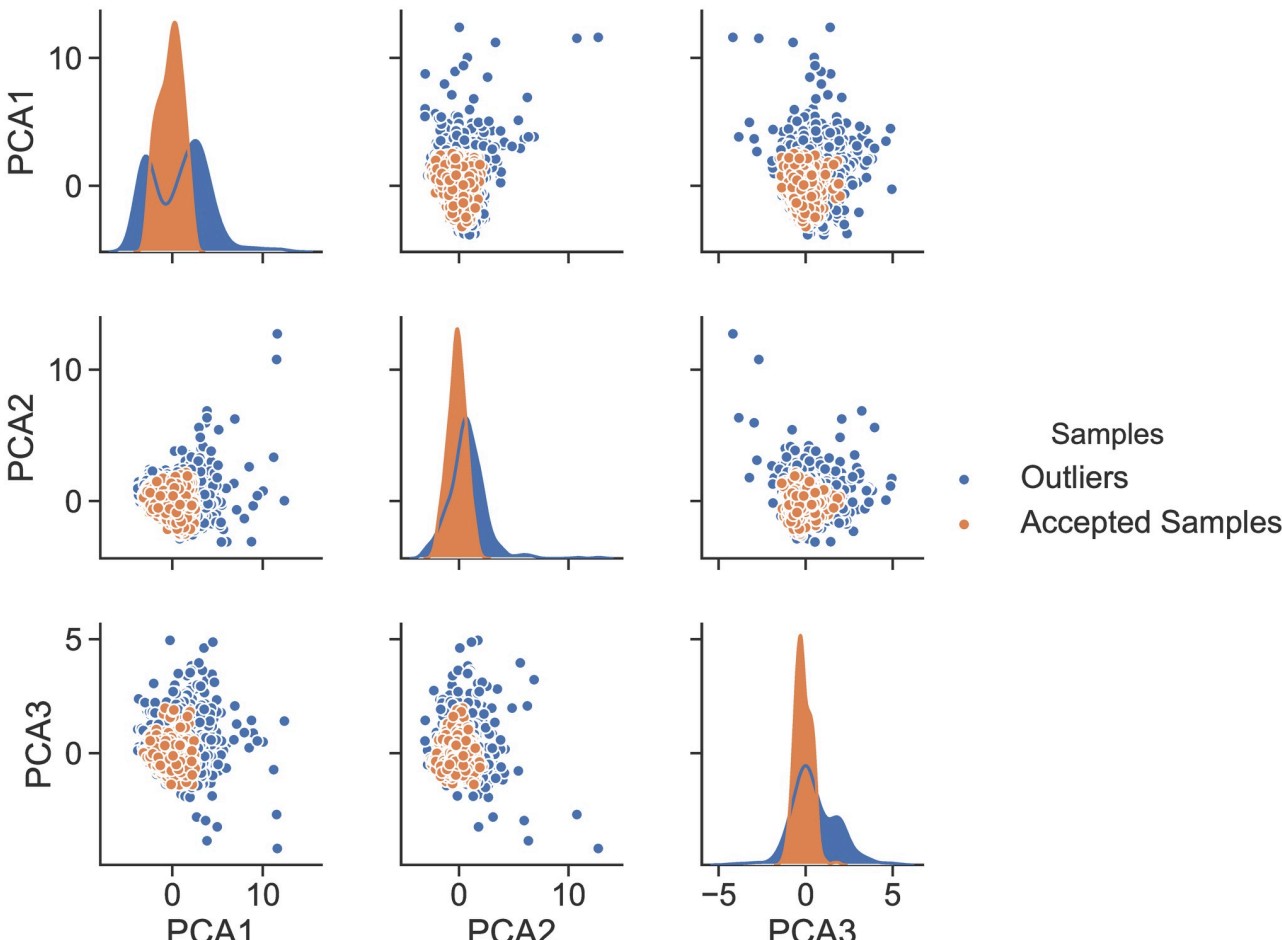

**Fig 1. Correlation plot of the buildings samples after applying PCA for dimension reduction, with outliers highlighted in blue.**

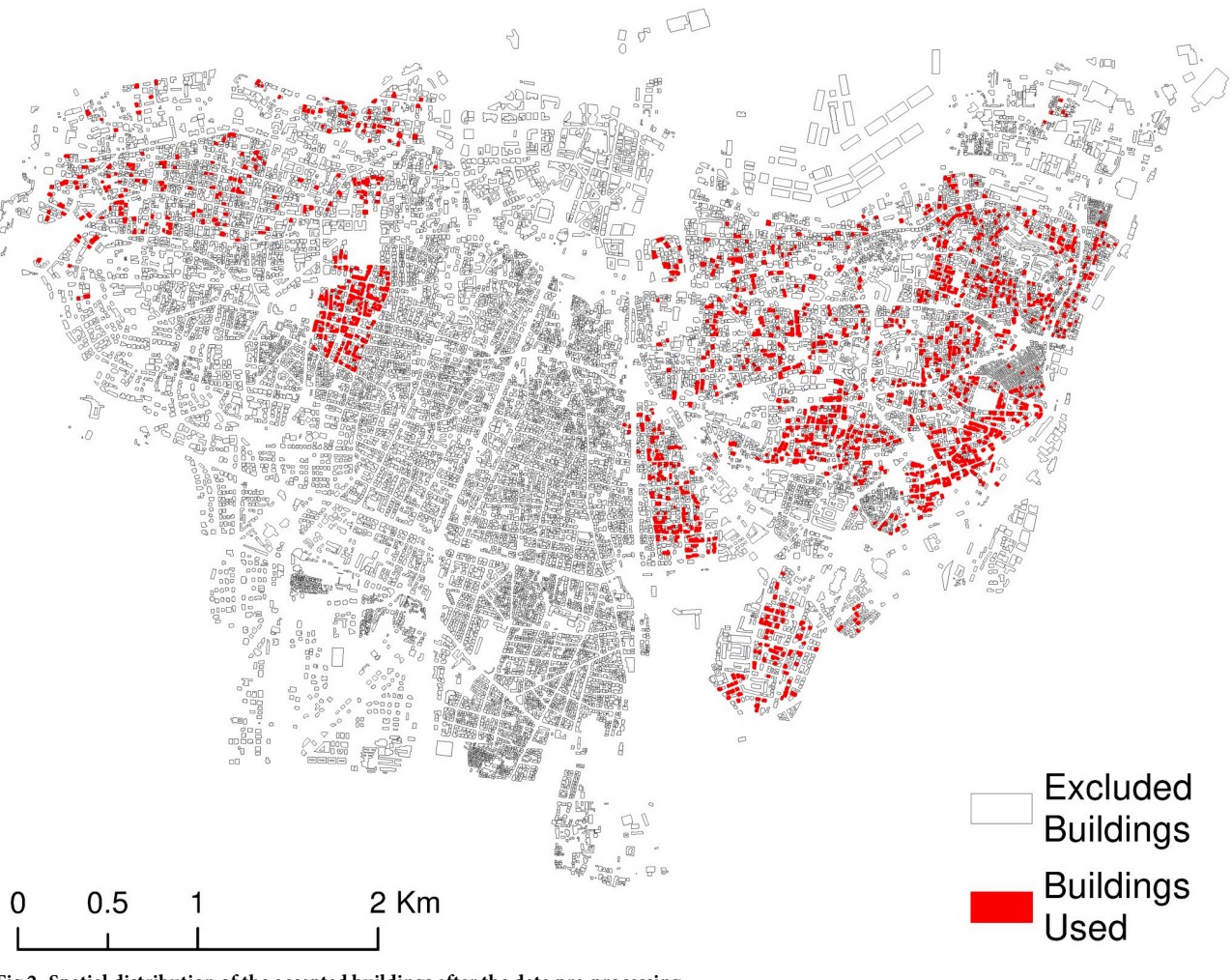

**Fig 2. Spatial distribution of the accepted buildings after the data pre-processing.**

## Methods

Many machine learning algorithms are available with different architectures. In our manuscript, we chose three well-known algorithms (linear and logistic regression, NN, and RF) that are described in more details below. Each model with a given architecture learns its parameters based on the training set. After that, to evaluate the model's performance, and thus to choose the best among them, a performance metric is applied to compare how close the actual data is to the model's prediction. This metric is normally a measure of error, or how far the predictions are from the actual data and thus the algorithm with the best metric value is chosen.

The building' floor number was shown to be dependent on the building's height, area, perimeter, and annual electricity consumption (Table 2). Having established this dependency, the building's construction period's relation to the aforementioned features was investigated. The selection of the buildings' features, that is the independent variables, can be justified by a correlation analysis achieved with the Pearson coefficient for the floors' number, which is used to evaluate bivariate correlation between continuous variables, and a dependency strength achieved with the logistic regression's accuracy score for the construction period, which is used to assess the accuracy of multi-label categorical classification as seen in Table 2. The

**Table 2. Pearson coefficient and logistic regression's accuracy score describing, respectively, the correlation and dependency between dependent variables and selected variables for prediction.**

| | Pearson Coefficient | Accuracy score |
| --- | --- | --- |
| | Floor number | Construction period |
| Electricity consumption | 0.57 | 0.51 |
| Height | 0.95 | 0.59 |
| Area | 0.37 | 0.46 |
| Perimeter | 0.38 | 0.46 |

Pearson coefficient is defined to be the ratio between the covariance between variables over the product of their respective variances given by $cov(x, y)/\sigma_x\,\sigma_y$. The accuracy score is given by: $accuracy = (1/n_{samples}) \times \Sigma_{i=1}^{n_{samples}} 1(y_{pred,i} = y_{true,i})$, where $y_{pred,i}$ is the predicted value of the *i-th* sample and $y_{true,i}$ is the corresponding true value.

For the prediction of number of floors, which is an integer value, a multi-layer feedforward (MLF) neural network (NN) for a multivalued non-linear regression was trained, whereas for the classification of the construction periods, which is a categorical value with labels ranging from 1 to 5 given in Table 1, NN, random forests (RF), and multiple logistic regression with classes from 1 to 5 were applied. MLF neural networks are the most popular type of NN. Their design is motivated from a real brain: networks of simple processing elements, neurons, operating on their local input data and communicating the output with other elements. Each neuron is connected to at least one other neuron, and each connection is evaluated by a weight coefficient. The training of a NN is in fact adjusting these weights in such way, the calculated outputs of the whole network are as close as possible to the actual ones [28]. RF are an ensemble learning method for classification or regression, which consist of constructing several estimators or decision trees at the training time and outputting the majority vote of the estimators for class prediction, or their mean prediction for regression [29]. Finally, multiple logistic regression is a classification method that describes the relationship between a nominal-scaled, i.e categorical variable and a set of independent variables. It consists of calculating the probabilities of the different possible outcomes of the categorical variable [30].

The number of hidden layers of the NN, which outputs the number of floors, ranged from 1 to 3, with corresponding number of neurons ranging from 3 to 8, and learning rate from 0.001 to 0.1. As for the construction period's logistic regression algorithm we used multiple solver to guarantee convergence such as Newton and BFGS solvers, a one-vs-the rest (OVR) multi-class strategy which consists of fitting one classifier per class, and finally features were selected according their k-score which is an inter-reliabilty measure for categorical variables. As for its NN, the hidden layers ranged 3 from to 8, with corresponding number of neurons varying from 1 to 40. The solvers we used were ADAM, BFGS, and Sigmoid, and a variety of activation functions were applied such as logistic, tanh, and relu. The learning rate was varied between 0.001 to 0.1. Finally, the RF estimators ranged from 10 to 500, with maximum depth ranging between 3 and 6, and maximum features used when considering the optimal split were defined using auto, and the criteria to evaluate the quality f the split was measure by the Gini impurity and the entropy. In order to measure the performance of the NN with different architectures, regression metrics such as the mean absolute error (MAE), the mean squared error (MSE), the mean absolute percentage error (MAPE), and the coefficient of determination ($R^2$) were computed. Similarly measures of performance of classification algorithms were also computed such as the accuracy score and the f1-score. Further, the resulting models were applied on the test sets to evaluate their performance. Subsequently, the best performing models were extended to the whole city.

## Floors

The dataset of 1, 536 samples was subdivided into training, validation, and test sets each containing respectively 859, 369, and 308 samples, which correspond respectively to the 55%, 25%, and 20% splits, often recommended in the literature [31]. The features of the training set were normalized and consequently their values ranged from 0 to 1. The NN's hyper-parameter tuning was carried out exploring different numbers of hidden units, neurons, and learning rates. Additionally, the cumulative distributions of the number of floors $P(f)$ of the 1,536 buildings and that of the combination of the latter set with the predicted 6,877 buildings' floors were computed. We tested whether these distributions can be explained by power-laws: $P(f) = (f/f_{min})^{-\alpha+1}$, where $f$ is the floor number, $\alpha$ is the exponent, and $f_{min}$ is the cutoff of the power-law, or whether a lognormal, whose parameters are given in [32], can better explains the distributions. The parameters were determined using the *poweRlaw* package in *R* for a discrete data set, bootstrapped, and subsequently the models were compared using the likelihood ratio test.

## Period of construction

Table 1 shows that the dataset is highly imbalanced, with only 1.2% of the buildings belonging to the first construction period, compared to 42.1% belonging to the third period. Resampling the data was crucial before proceeding. Since we had a relatively small dataset (1, 536 samples), oversampling the minority classes of the training set was applied to improve the quality of the predictive model. This was achieved using SVMSMOTE [33] by creating synthetic observations of the minority classes, at each iteration of the cross-validation. Different configurations of logistic regression, RF classifiers and NN classifiers were examined and compared with the accuracy score.

# Results

## Floors prediction

The input's layer's 4 neurons correspond to the area, perimeter, height, and electricity consumption, while to output layer's single neuron is that of the period of construction. The optimum number of hidden layers and their neurons were found to be 1 and 8 respectively, with a learning rate of 0.01 and a sigmoid transfer function. The scores of the applied NN on the test are given by:

- mean absolute error MAE = 0.54

- mean squared error MSE = 0.73

- mean absolute percentage error MAPE = 7.2%

- $R^2$ = 87.7%

The prediction of the floors' number for the rest of the city's buildings could now be extended keeping in mind that buildings with missing input features had to be excluded. This left us with 6, 877 buildings whose number of floors is to be predicted. The results along with the surveyed data from USJ were mapped as shown in Fig 3.

The cumulative distribution of the number of floors of the USJ buildings was evaluated. Additionally, the latter along with predicted buildings' floor number was also computed. They are shown respectively in Figs 4 and 5. In the first, the ratio $r$ of the log-likelihoods of the data between the power-law and lognormal is negative, which means that the lognormal is a better fit, while in the second $r > 0$ indicating that the power law with exponent $\alpha$ = 5.35±0.30 is a

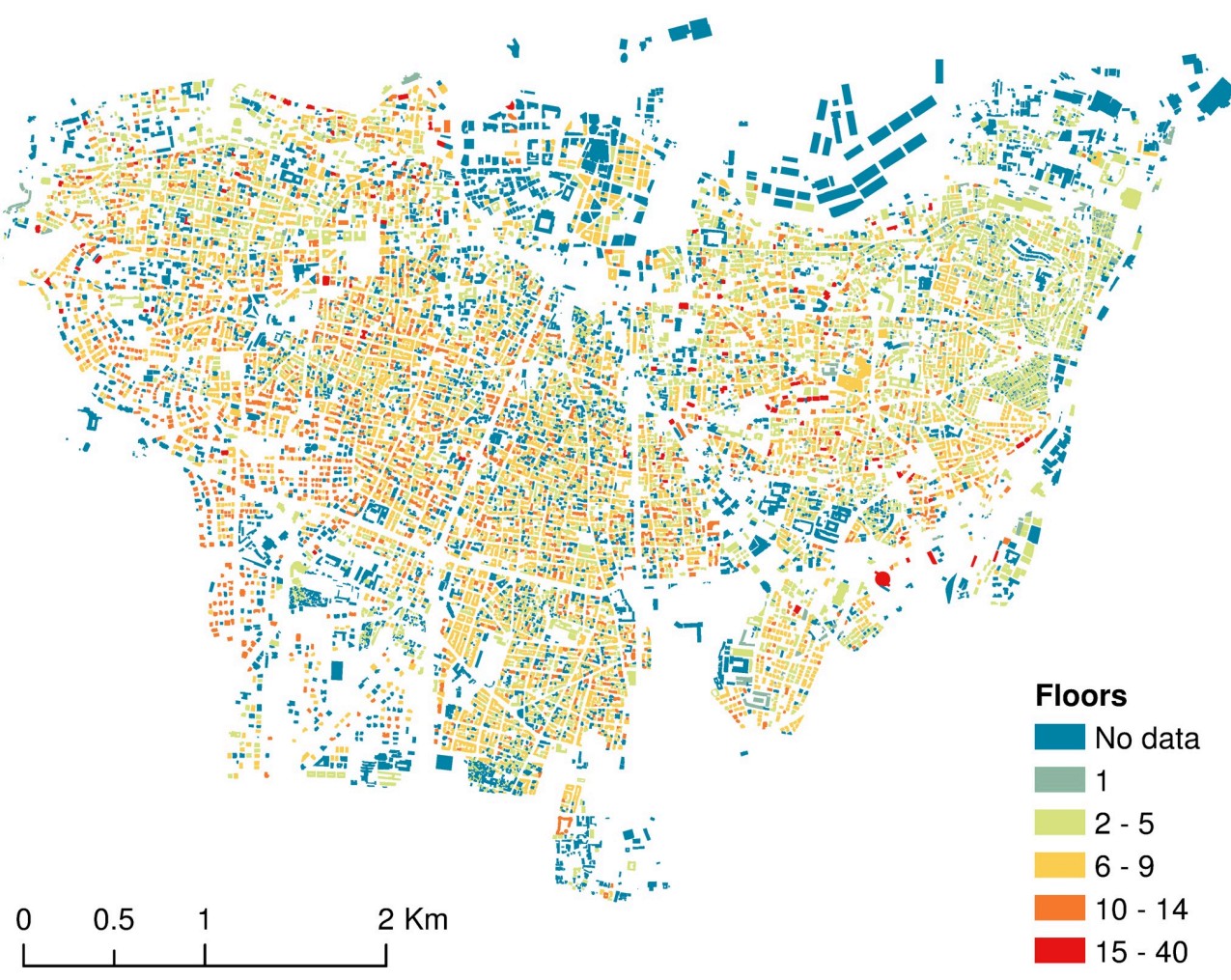

**Fig 3. Distribution of buildings per floors' number in Beirut administrative area.**

better fit. This latter parameter is in accordance with the findings of [34], where the exponent of the height distribution of London was shown to be $\alpha = 5.26$.

### Period of construction prediction

The exhaustive hyper-parameter tuning and models cross-comparison converged to a random forest with 100 decision trees with an accuracy score of 56.7%. Further, its accuracy score on the test set was given by 48.7%. Using this model, despite its low accuracy, the rest of Beirut's buildings were tagged with their corresponding predicted construction period (Fig 6). Further, the confusion matrix was plotted in Fig 7. It revealed that the algorithm best predicted the third construction period with an accuracy of 63%, while its worse accuracy was attained with the second construction period with only 40%.

The sensitivity of the pipeline to our desired methodology was also tested. Here we present an illustration of the effect of sampling and normalization on RF. It is worth noting that without sampling the model misses all of the buildings from the first construction period as shown in Table 3.

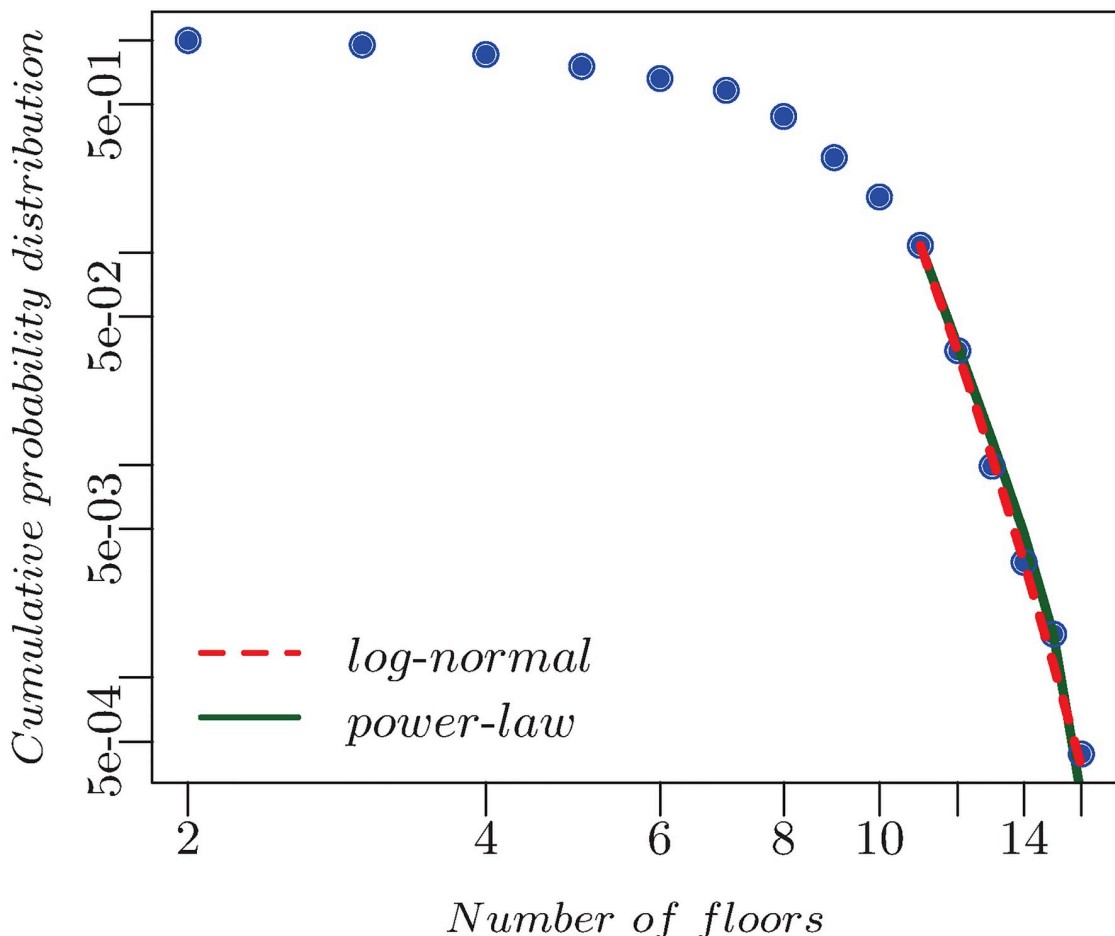

**Fig 4. $P(f)$ of the 1,536 building buildings is shown blue, the power-law is shown in green, and the lognormal in shown in red.**
Their respective parameters are $f_{min} = 11$ and $\alpha = 12.92$, while those of the lognormal are given by $f_{min} = 11$, $\mu = 1.55$, and $\sigma = 0.27$.

## Discussion

In the previous sections, we have presented a pipeline which relies on machine learning to complement urban buildings' database. It should be noted that our starting point was a data whose floors distribution is best described by a lognormal. However the combination of the latter with the predicted data was shown to be a power-law, which is in full accordance with the measured one for other cities [34]. The fact that the distribution of predicted buildings' heights follows a power-law and not a log-normal is a confirmation that our model recovers known properties about the heights; namely that they follow a power-law and not a log-normal distribution. This is further a consistency check on the validity of the results. High accuracy in predicting buildings' floors number was attained revealing a strong relation with the building height, area, perimeter and electricity consumption. The quantification of the floors is relevant to energy planning, as it helps simulating the energy demand by representing each floor as a thermal zone. A floor can be further subdivided into subzones for more accuracy of the building performance simulation [35]. Furthermore, it can help approximating the building population for micro-scale modeling and analysis of human behavior [36].

On the other hand, the period of construction could be predicted with an accuracy score of 48.7% only. More training data may be required. However, the low accuracy may be related to

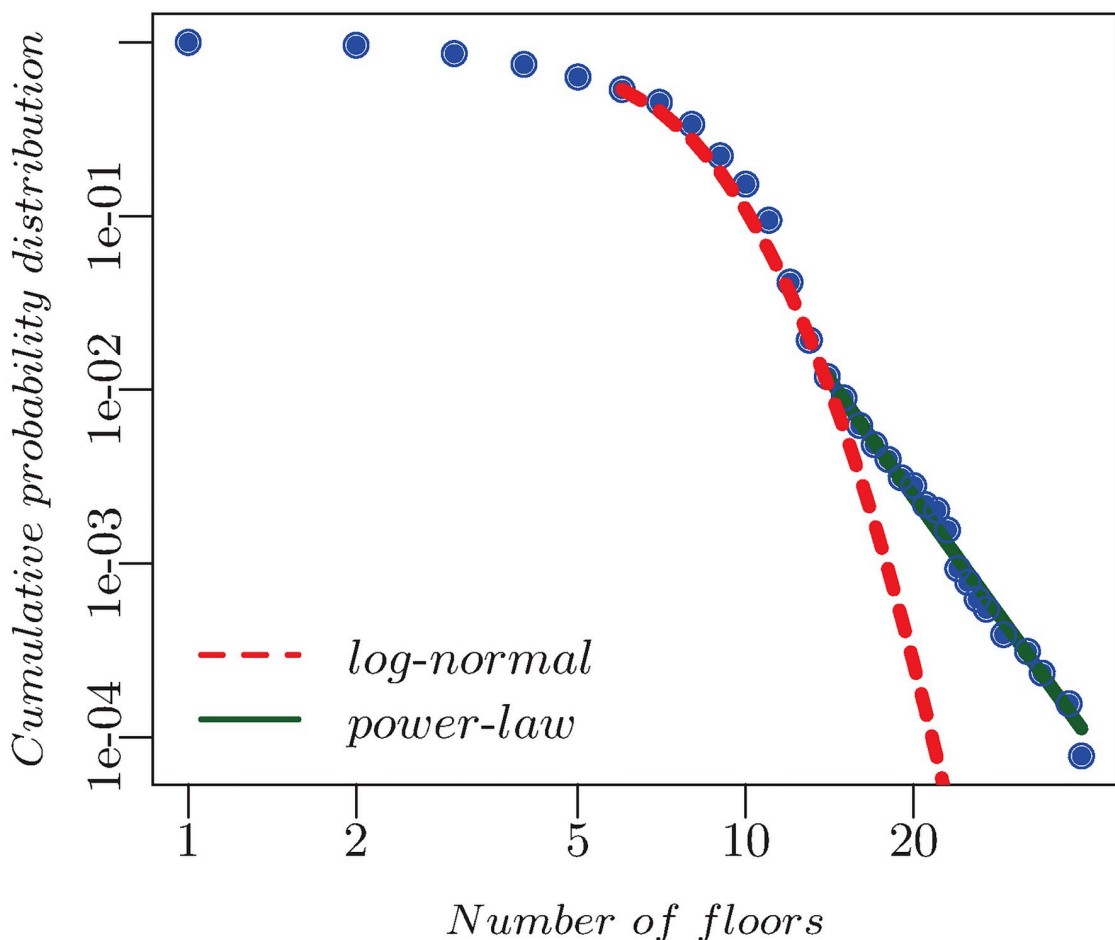

**Fig 5. $P(f)$ for all the buildings is shown in blue.** The green line is the power-law with $f_{min} = 14$, and $\alpha = 5.35$. while the red line corresponds to the lognormal with $f_{min} = 6$ params = 2.05, 0.28.

the need for more variables on which the period of construction depends, such as window to wall ratio (WWR), wall thickness and other era-specific descriptors. The construction period gives insight into the materials of the buildings, which can inform materials flows and stocks models for valuation of buildings, as well as the determination of their energy performance and refurbishment techniques [37, 38], and the identification of future waste streams along with recovery strategies [39].

## Conclusion

Finally, we developed NN algorithms to predict the number of floors and the construction period of buildings given their heights, areas, perimeters and electricity consumption. We began by cleaning the available dataset and removing unreliable entries and outliers. Then, we evaluated the significance of each input feature on the output to justify its selection. The NN was able to predict the number of floors with a high prediction accuracy with a coefficient of determination of $R^2$ of 87.7%. Then, the construction period's Random Forest was built after re-sampling of the data to overcome its imbalance. Subsequently, the exponent of the power-law governing the floor distribution was shown to be conforming with that appearing in the literature.

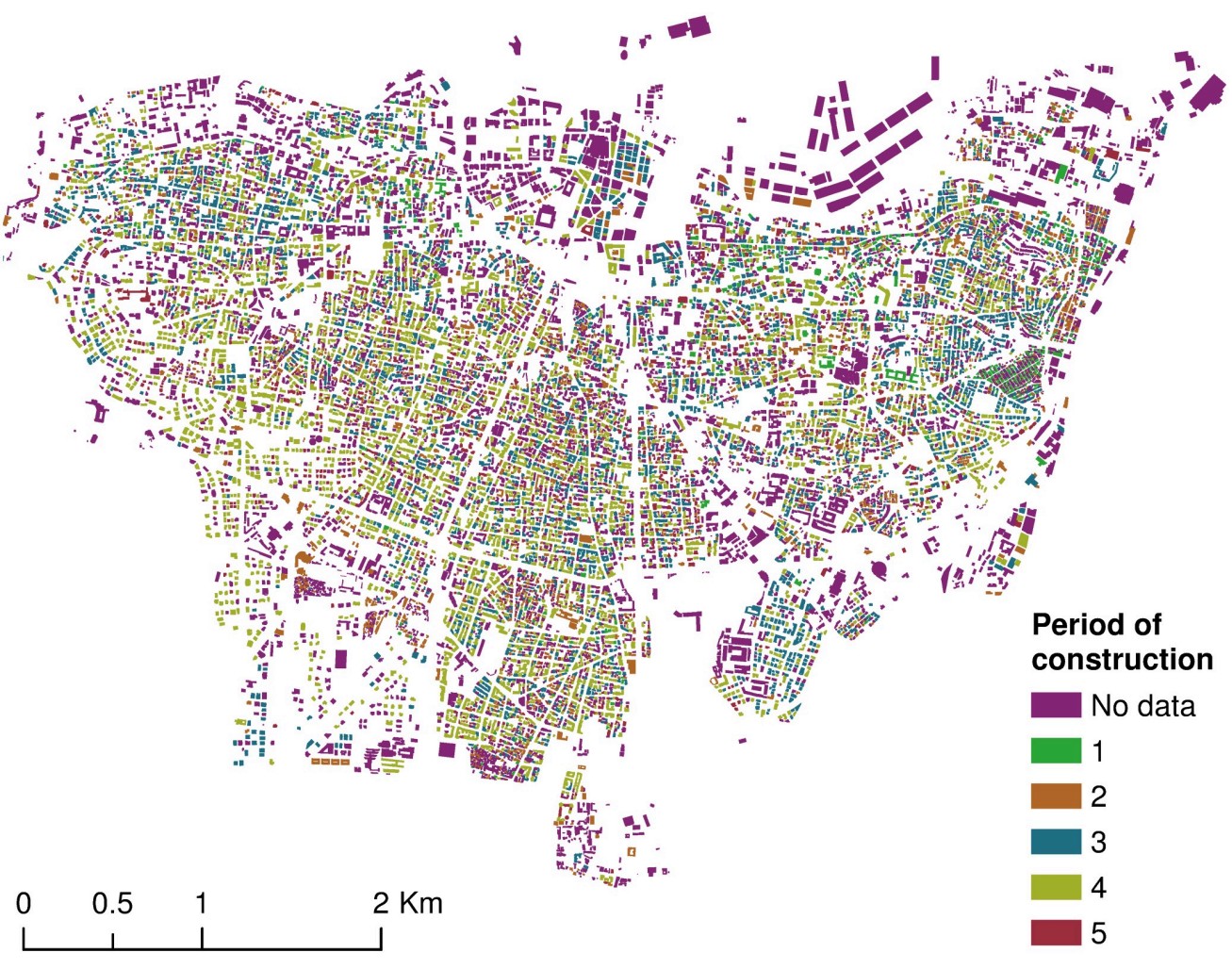

**Fig 6. Spatial distribution of buildings per period of construction in Beirut administrative area.**

In developing cities, like Beirut, available urban data is often underutilized because of its sporadic nature and/or access challenges. With the approach adopted in this paper, the lack of full datasets is compensated by machine learning interventions that can fill in data gaps and offer policy designers a powerful and verifiable new leverage. Beside the immediate applications reported above related to service provision, efficiency, and analytics (e.g. electricity) and buildings' characteristics, the presented methodology can be an effective tool to generate wider policy insights despite data irregularities. The two main areas where such an approach can be particularly advantageous are: (1) assessing urban resiliency, risk, and emergency planning. For example, having an accurate distribution of the number of floors and building materials would be critical for a rapid assessment of the human loss in the case of a natural disaster such as an earthquake or large fires; (2) generating demographic and socio-economic insights related to population concentration, census, which is not available in Lebanon since 1994, and public services. For example, number of floors distribution could be used to distinguish between residential, commercial, and industrial units/zones within the city and inform policy experts about electricity rationing strategy (like the case of Beirut where power outages are regular but randomly allocated geographically); or provide information on energy consumption's

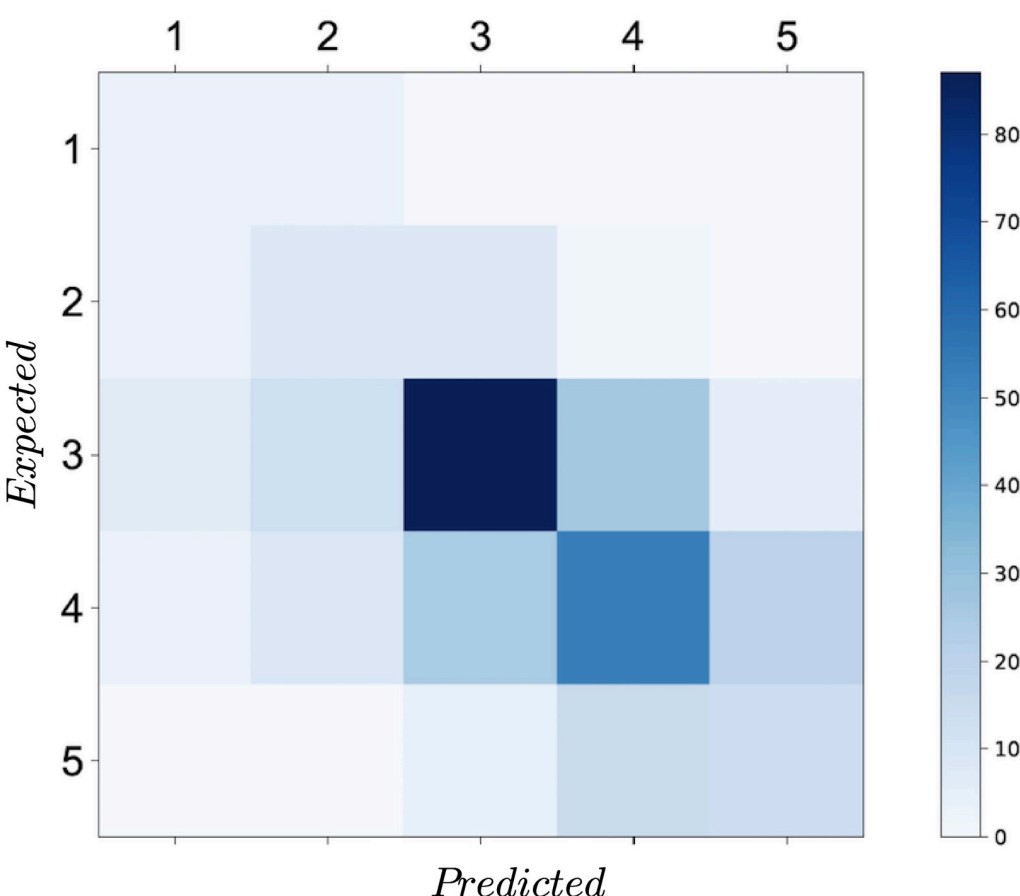

**Fig 7. Distribution of buildings per predicted period of construction in Beirut administrative area, with colorbar depicting the number of floors.**

**Table 3. Sample of the pipeline's sensitivity analysis.**

| Construction period | RF (sampling only) | RF (sampling and normalization) |
|---|---|---|
| 1 | 0% | 37.5% |
| 2 | 30.9% | 43.6% |
| 3 | 62.1% | 51.5% |
| 4 | 46.1% | 44.7% |
| 5 | 53.1% | 46.5% |

"hot spots" which could help with predicting electricity demand surge and the needed grid reinforcement strategy.

## Acknowledgments

We would like to thank Dr. Ali Ahmad for the useful discussion and feedback on the manuscript.

## Author Contributions

**Conceptualization:** Sara Najem.

**Data curation:** Alaa Krayem, Sara Najem.

**Formal analysis:** Ghaleb Faour, Sara Najem.

**Methodology:** Alaa Krayem, Aram Yeretzian, Sara Najem.

**Writing – original draft:** Alaa Krayem, Aram Yeretzian, Ghaleb Faour, Sara Najem.

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
