## [Decision Letter · Decision Letter 0]

8 Jul 2020

PONE-D-20-10968

Machine learning for buildings' characterization and

power-law recovery of urban metrics

PLOS ONE

Dear Dr. Najem,

Thank you for submitting your manuscript to PLOS ONE. After careful consideration, we feel that it has merit but does not fully meet PLOS ONE’s publication criteria as it currently stands. Therefore, we invite you to submit a revised version of the manuscript that addresses the points raised during the review process.

The paper is very exciting. However, as reviewer 1 says it clearly:  the manuscript in its current form is not suitable for publication in an interdisciplinary journal like PLOS One, as it is currently located in a space where it has not enough detail for subject experts (e.g.  what kind of NN model did you use?) and not explanatory enough for non-experts (e.g. what does it mean, if the results fit one distribution better than another?). Both reviewers give you insights to improve it and to bring it more into a form that is suitable for this journal as it is in its core very interesting work, about which one would like  to know more details about.

We look forward to receiving your revised manuscript.

Kind regards,

Celine Rozenblat

Academic Editor

PLOS ONE

Additional Editor Comments:

As reviewer 1 says it clearly: the manuscript in its current form is not suitable for publication in an interdisciplinary journal like PLOS One, as it is currently located in a space where it has not enough detail for subject experts (e.g. what kind of NN model did you use?) and not explanatory enough for non-experts (e.g. what does it mean, if the results fit one distribution better than another?). Both reviewers give you insights to improve it and to bring it more into a form that is suitable for this journal as it is in its core very interesting work, about which one would like to know more details about.

2. There are a number of broken figure references, e.g. line 69, 134.

Please ensure these are fixed in the revised version of the manuscript.

In addition, please update your data availability statement to give a full list of data sources and URL links or contact details that future researchers can use to access the data.

3. We note that you have stated that you will provide repository information for your data at acceptance. Should your manuscript be accepted for publication, we will hold it until you provide the relevant accession numbers or DOIs necessary to access your data. If you wish to make changes to your Data Availability statement, please describe these changes in your cover letter and we will update your Data Availability statement to reflect the information you provide

Reviewers' comments:

Reviewer's Responses to Questions

**Comments to the Author**

1. Is the manuscript technically sound, and do the data support the conclusions?

Reviewer #1: Yes

Reviewer #2: Yes

2. Has the statistical analysis been performed appropriately and rigorously? 

Reviewer #1: Yes

Reviewer #2: Yes

3. Have the authors made all data underlying the findings in their manuscript fully available?

Reviewer #1: No

Reviewer #2: No

4. Is the manuscript presented in an intelligible fashion and written in standard English?

Reviewer #1: Yes

Reviewer #2: Yes

5. Review Comments to the Author

Reviewer #1: This work estimates additional properties of buildings in Beirut, Lebanon. On the basis of a smaller data set, the authors use methods from machine learning to estimate the number of floors and year of construction. The resulting data - given a certain validity of the method - provides novel insights into the building stock of the city, which is crucial in developing cities such as Beirut. It might be a valuable tool for a resilient future development in a place where no recent census data exists, or (building) information is scarce. I think this work is of interest for a wide community of scholars, as it shows some of the potential machine learning methods have when data accessibility and availability is limited, as it is often the case in poorer countries.

Although the manuscript appeals to me, I think a few clarifications and improvements are necessary. It seems to me, that the data cleaning process is a little too stringent, as more than two thirds of the buildings get removed. Does the USJ data set not represent a good sample of the building stock in Beirut? I think it would improve this part, if a little discussion about the data would be there, or a table/figure that shows the distribution of the two used parameters for the full data set. Especially, since so many buildings are removed already in this first step.

I have always a hard time reading 3D plots in manuscripts. I think it might be more informative if Figure 1 was similar to a correlation plot (or sometimes called 'scatter plot matrix'). Also, I would find it informative, to see what the Principle components look like, as they can tell a lot about how the data looks like in general.

I think it would be good to make the methods section a little bit more understandable for non-experts, as PLOS One has a readership from across different fields. For example, that it would be great if the specific choices for the different algorithms would be explained in a little more detail. Also, a non-expert might not immediately know what the meaning of the different scores exactly is and what information they exactly provide to the precision of the methods. The same accounts for why the authors chose the specific subsets sizes to train, validate, and test the model. I believe that the whole Methods section would benefit from such additional explanations.

This extends into the results section, where I would love to see the different results from the exploration of different models, number of layers, non-normalized vs. normalized data, and so on.

In general, it would be interesting to see how sensitive the pipeline is to changes and what the different results were during the exploration step. As this might be crucial if other people would want to use the same method.

The authors compare the distribution of floors in Beirut to a power-law and a lognormal distribtution. What does it mean that they follow more one or the other? What are the additional insights one gains from this?

I have mentioned before, the work is very appealing for me. However, I think the manuscript in its current form is not suitable for publication in an interdisciplinary journal like PLOS One, as it is currently located in a space where it has not enough detail for subject experts (e.g. what kind of NN model did you use?) and not explanatory enough for non-experts (e.g. what does it mean, if the results fit one distribution better than another?). I advocate for some major revisions to bring it more into a form that is suitable for this journal as it is in its core very interesting work, about which I want to know more detail about.

Reviewer #2: “Machine learning for buildings? characterization and power-law recovery of urban metrics”

referee-report

The authors analyze building data of Beirut in Lebanon with the purpose of predicting building age and the number of floors. Specifically, a somewhat small subset of surveyed buildings is considered. As “independent variables” height, area, perimeter, and electricity consumption are used and fed into the neural networks. The work achieves good performance for the number of floors and modest performance for the period construction. The authors complement an analysis of the distribution of number of floors and find somewhat large exponents beyond 5.

The paper is well written and the approach can be of importance for similar applications in other cities and countries. The need for building data is well justified in the introduction. I appreciate that the manuscript is short.

I have a few issues that the authors should address:

- Please clarify if the city of Beirut or the metro-region is considered. I guess it is the former. Please also add an approximate population figure. Dividing the population by the number of buildings gives a rough idea about population density/floors.

- In my opinion 3D representation inadequate never works in 2D. Please develop an alternative representation.

- The power-law exponent is very large (also in the publication by Batty). The problem is that such steep power-law distributions loose what makes power-laws special and they become similar to other distributions.

- The prediction of period of construction could be improved by including information on location, e.g. distance from center.

6. PLOS authors have the option to publish the peer review history of their article (what does this mean?). If published, this will include your full peer review and any attached files.

Reviewer #1: No

Reviewer #2: No

---

## [Author Response · Author response to Decision Letter 0]

26 Aug 2020

We thank the Editor for seeing merit in our work and for sending us the Referees’ reports. These provided valuable input and recommendations. We also appreciate that both Referees saw value in our work “I think this work is of interest for a wide community of scholars,” (quoting the first Referee) and “The paper is well written and the approach can be of importance for similar applications in other cities and countries.” as the second Referee states. We particularly appreciate the Referees’ helpful suggestions on technical points and details of presentation, which will ensure our paper is more easily accessible to a broad audience.

As we were preparing for the reply, a devastating blast hit Beirut, and took us all by surprise. We see the urgency of putting this work out to inform the scientific community about the buildings’ details, which are part of all the modeling initiatives calling for the dissemination of such data: be it the shock wave simulation in this complex urban environment, damage assessment, ret- rograding and buildings’ preservation. We also provide a link to our dataset: https://zenodo.org/record/4001720#.X0ZaeZMzblw

Concerning the detailed comments, we address them below and list the corresponding changes in the manuscript. Original comments of the Referees are in blue and changes in the manuscript are in red, both here and in the revised manuscript.

With these changes and clarification, we trust our manuscript is now suitable for publication in PLOS One.

Reply to the first Referee:

Referee 1: This work estimates additional properties of buildings in Beirut, Lebanon. On the basis of a smaller data set, the authors use methods from machine learning to estimate the number of floors and year of construction. The resulting data - given a certain validity of the method - provides novel insights into the building stock of the city, which is crucial in developing cities such as Beirut. It might be a valuable tool for a resilient future development in a place where no recent census data exists, or (building) information is scarce. I think this work is of interest for a wide community of scholars, as it shows some of the potential machine learning methods have when data accessibility and availability is limited, as it is often the case in poorer countries. Although the manuscript appeals to me, I think a few clarifications and improvements are necessary. It seems to me, that the data cleaning process is a little too stringent, as more than two thirds of the buildings get removed. Does the USJ data set not represent a good sample of the building stock in Beirut? I think it would improve this part, if a little discussion about the data would be there, or a table/figure that shows the distribution of the two used parameters for the full data set. Especially, since so many buildings are removed already in this first step.

Reply: We thank the Referee for characterizing our work as novel, crucial, and valuable. We also acknowledge that their reservations on the current manuscript’s methodology section, given that readership of PLOS One, are very valid.

Indeed, as the Referee notes, in the data cleaning process more than two thirds of the buildings are filtered out. The USJ data set is a very good representative one, however it did not include the building’s height as a descriptor. Buildings’ footprints were manually digitized over the entire city of Beirut by CNRS-L using aerial photos at 15cm resolution and VHR pan-chromatic satellite images from Pleaides-1A at 70cm resolution. This is not part of our work but rather a CNRS-L in-house processing step, which was made available to us. A large number of these buildings, when overlaid on the USJ dataset, turned out to have atypical floor height (≤ 2.8m or ≥ 4, 5m) and thus were removed from the dataset. This lead to the filtering of around two-thirds of the buildings.

Action: This sentence was added to the text: It is worth noting that buildings’ footprints were manually digitized over the entire city of Beirut by CNRS-L using aerial photos at 15cm resolution and VHR pan-chromatic satellite images from Pleaides-1A at 70cm resolution. This is not part of our work but rather a CNRS-L in-house processing step, which was made available to us.

Referee 1: I have always a hard time reading 3D plots in manuscripts. I think it might be more informative if Figure 1 was similar to a correlation plot (or sometimes called ’scatter plot matrix’). Also, I would find it informative, to see what the Principle components look like, as they can tell a lot about how the data looks like in general.

Reply: We thank the Referee for his/her suggestion to use a different representation for the PCA. We added the below scatter plot matrix in addition to the 3D plot in our original document as they carry the same information in different dimensions.

Action: The below figure was added along with the sentence “To visualize the outliers, the six features were reduced to three using the Principle Component Analysis (PCA), which allowed for a 3D representation of the samples as function of the new dimensions, as shown in Fig. 1 as well as the corresponding two-dimensional scatter plot matrix shown in Fig. 2 . ”

2

 Correlation plot of the buildings samples after applying PCA for dimension reduction, with outliers highlighted in brown.

Referee 1: I think it would be good to make the methods section a little bit more understandable for non-experts, as PLOS One has a readership from across different fields. For example, that it would be great if the specific choices for the different algorithms would be explained in a little more detail. Also, a non-expert might not immediately know what the meaning of the different scores exactly is and what information they exactly provide to the precision of the methods. The same accounts for why the authors chose the specific subsets sizes to train, validate, and test the model. I believe that the whole Methods section would benefit from such additional explanations. This extends into the results section, where I would love to see the different results from the exploration of different models, number of layers, non-normalized vs. normalized data, and so on.

Reply: We agree with the Referee that the algorithms and the scores need to be defined properly to justify their use. As for the sizes of the training, validation, and test sets these correspond to the 55%, 25%, and 20% respectively often recommended in the literature. For the results section, we presented the optimal combination of parameters for the different models which minimized the error measures.

Action: The following text in red was added to the beginning of the Methods section: “The selection of the buildings’ fea-

tures, that is the independent variables, can be justified by a correlation analysis achieved with the Pearson coefficient for

the floors’ number, which is used to evaluate bivariate correlation between continuous variables, and a dependency strength

achieved with the logistic regression’s accuracy score for the construction period, which is used to assess the accuracy of multi-

label categorical classification as seen in Table 2. The Pearson coefficient is defined to be the ratio between the covariance

between variables over the product of their respective variances given by cov(x,y)/σxσy. The accuracy score is given by:

accuracy = (1/nsamples) × Σnsamples 1(ypred,i = ytrue,i), where ypred,i is the predicted value of the i − th sample and ytrue,i i=1

is the corresponding true value.”

We also justify the split in the sizes of the training, validation, and test sets by adding the following: “The dataset of 1, 536 samples was subdivided into training, validation, and test sets each containing respectively 859, 369, and 308 samples, which correspond respectively to the 55%, 25%, and 20% splits, often recommended in the literature.

The following was added to give all the details about the algorithms used: “MLF neural networks are the most popular type of NN. Their design is motivated from a real brain: networks of simple processing elements, neurons, operating on their local input data and communicating the output with other elements. Each neuron is connected to at least one other neuron, and each connection is evaluated by a weight coefficient.The training of a NN is in fact adjusting these weights in such way,

the calculated outputs of the whole network are as close as possible to the actual ones [37]. RF are an ensemble learning method for classification or regression, which consist of constructing several estimators or decision trees at the training time and outputting the majority vote of the estimators for class prediction, or their mean prediction for regression [36]. Finally, multiple logistic regression is a classification method that describes the relationship between a nominal-scaled , i.e categorical variable and a set of independent variables. It consists of calculating the probabilities of the different possible outcomes of the categorical variable [38]. The number of hidden layers of the NN, which outputs the number of floors, ranged from 1 to 3, with corresponding number of neurons ranging from 3 to 8, and learning rate from 0.001 to 0.1. As for the construction period’s logistic regression algorithm we used multiple solver to guarantee convergence such as Newton and BFGS solvers, a one-vs-the rest (OVR) multi-class strategy which consists of fitting one classifier per class, and finally features were selected according their k-score which is an inter-reliabilty measure for categorical variables. As for its NN, the hidden layers ranged 3 from to 8, with corresponding number of neurons varying from 1 to 40. The solvers we used were ADAM, BFGS, and Sigmoid, and a variety of activation functions were applied such as logistic, tanh, and relu. The learning rate was varied between 0.001 to 0.1. Finally, the RF estimators ranged from 10 to 500, with maximum depth ranging between 3 and 6, and maximum features used when considering the optimal split were defined using auto, and the criteria to evaluate the quality f the split was measure by the Gini impurity and the entropy.”

Referee 1: In general, it would be interesting to see how sensitive the pipeline is to changes and what the different results were during the exploration step. As this might be crucial if other people would want to use the same method.

Reply: We Thank the Referee for asking this important question on the sensitivity of the pipeline. We reassert that the reported models outperformed all the others and here we show a sample of how sensitive the prediction period to the normalization and sampling.

TABLE I. Sample of the pipeline’s sensitivity analysis.

Construction period RF with sampling without normalization RF with sampling and normalization

3

 1 0%

2 30.9%

3 62.1%

4 46.1%

5 53.1%

37.5% 43.6% 51.5% 44.7% 46.5%

 Action: The above table was added to the text. The sensitivity of the pipeline to the our desired methodology was also tested. Here 191 we present an illustration of the effect of sampling and normalization on RF. It is worth 192 noting that without sampling the model misses all of the buildings from the first 193 construction period as shown in Table 3.

Referee 1: The authors compare the distribution of floors in Beirut to a power-law and a lognormal distribtution. What does it mean that they follow more one or the other? What are the additional insights one gains from this?

Reply/Action: We added the following sentences in red to clarify the meaning of this finding: The fact that the distribution of predicted buildings’ heights follows a power-law and not a log-normal is a confirmation that our model recovers known properties about the heights. This is further a consistency check on the validity of the results. These distributions are namely indicators of the underlying dynamical processes that generate them: power-laws result from multiplicative processes while log-normal from additive log-Gaussian ones [28].

Referee 1: I have mentioned before, the work is very appealing for me. However, I think the manuscript in its current form is not suitable for publication in an interdisciplinary journal like PLOS One, as it is currently located in a space where it has not enough detail for subject experts (e.g. what kind of NN model did you use?) and not explanatory enough for non-experts (e.g. what does it mean, if the results fit one distribution better than another?). I advocate for some major revisions to bring it more into a form that is suitable for this journal as it is in its core very interesting work, about which I want to know more detail about.

Reply: We thank the Referee for stressing what what we need to direct our attention to in this revised version of the manuscript. These points have been addressed in details above.

Reply to the second Referee:

Referee 2: Please clarify if the city of Beirut or the metro-region is considered. I guess it is the former. Please also add an approximate population figure. Dividing the population by the number of buildings gives a rough idea about population density/floors.

Reply: As the Referee correctly points out our study area is the city of Beirut. The first Referee states one of the challenges of studying a city like Beirut is the absence of census data. The latest was carried out in 1994. Therefore, any population density estimation is flawed with errors. We argue that surveying

Action: This was added to clarify the area of our study: “Beirut, the city, is located on the eastern shore of the Mediterranean sea with a stock of 17, 742 buildings (in 2016).”

Referee 2: In my opinion 3D representation inadequate never works in 2D. Please develop an alternative representation. Reply: Both Referees agree about the representation of the PCA in 3D. The point the second Referee raises was addressed by

adding the 2D scatter plot matrix shown in Figure 2 in this revised version of the manuscipt.

Referee 2: The power-law exponent is very large (also in the publication by Batty). The problem is that such steep power-law distributions loose what makes power-laws special and they become similar to other distributions.

Reply: We agree with the Referee that the value of the power-law exponent is large. However, it could still be differentiated from a lognormal distribution through the log-likelihood ratio.

Referee 2: The prediction of period of construction could be improved by including information on location, e.g. distance from center.

Reply: We agree with the Referee that the location from the center could be a contributing factor in the prediction of the year of construction. It is worth noting that we have started our experimentation with the latitude and longitude as independent variables, which proved to be way less significant that the ones we kept. We also looked at correlation between buildings’ year of construction and their relative distance from the center. This we believe to be caused by the fact that our dataset is already not spatially very extended from the center and sparse when it comes to the distribution of years of construction. However, the point the Referee makes is a very important one and would give lead into the historical evolution of the city.

4

---

## [Decision Letter · Decision Letter 1]

26 Nov 2020

PONE-D-20-10968R1

Machine learning for buildings' characterization and

power-law recovery of urban metrics

PLOS ONE

Dear Dr. Najem,

Thank you for submitting your manuscript to PLOS ONE. After careful consideration, we feel that it has merit but does not fully meet PLOS ONE’s publication criteria as it currently stands. Therefore, we invite you to submit a revised version of the manuscript that addresses the points raised during the review process.

Thanks for having revised the article which is improved now. Please address the requests of reviewer 1...

We look forward to receiving your revised manuscript.

Kind regards,

Celine Rozenblat

Academic Editor

PLOS ONE

Reviewers' comments:

Reviewer's Responses to Questions

**Comments to the Author**

1. If the authors have adequately addressed your comments raised in a previous round of review and you feel that this manuscript is now acceptable for publication, you may indicate that here to bypass the “Comments to the Author” section, enter your conflict of interest statement in the “Confidential to Editor” section, and submit your "Accept" recommendation.

Reviewer #1: (No Response)

Reviewer #2: All comments have been addressed

2. Is the manuscript technically sound, and do the data support the conclusions?

Reviewer #1: Yes

Reviewer #2: Yes

3. Has the statistical analysis been performed appropriately and rigorously? 

Reviewer #1: Yes

Reviewer #2: Yes

4. Have the authors made all data underlying the findings in their manuscript fully available?

Reviewer #1: (No Response)

Reviewer #2: (No Response)

5. Is the manuscript presented in an intelligible fashion and written in standard English?

Reviewer #1: No

Reviewer #2: Yes

6. Review Comments to the Author

Reviewer #1: I thank the authors for taking my input into account. The manuscript has

largely been improved in my opinion. However, there are still a few minor

points I would like the authors to address:

1) The diagonal of the scatter plot matrix could be used to show the

distribution of the two classes. It would add valuable information to the

figure. An example of such a plot can be found at, for example,

https://seaborn.pydata.org/examples/scatterplot_matrix.html.

2) Since all information is contained in figure 2, I would recommend removing

figure 1, as it does not provide any useful additional information. But this I

leave to the authors.

3) It would be, additionally, be very informative to have a map similar to

figures 3 and 4 with the buildings that are actually used in the analysis.

4) Please provide a few references to the added part in lines 136-137 to

justify the percentages used.

5) In line 202 you mention that you recover known properties about heights.

Please add a sentence what these properties are or add references.

6) I apologize reiterating this point again, but in the case of buildings,

what does it mean that the underlying processes are either multiplicative or

additive? I personally have no intuition what that means in terms of building

heights. Please clarify further.

7) I appreciate that you added the section from line 100 and onward. However,

it does still not clarify why MLF-NNs are a good choice for the analysis you

did. I'm no expert in machine learning, so please make this point a little

clearer for people like me.

Reviewer #2: (No Response)

7. PLOS authors have the option to publish the peer review history of their article (what does this mean?). If published, this will include your full peer review and any attached files.

Reviewer #1: No

Reviewer #2: No

---

## [Author Response · Author response to Decision Letter 1]

2 Dec 2020

We thank the Editor for seeing merit in our work and for sending us the Referees’ reports. We also thank the Referees positive comments on the revised version of the manuscript, which is seen to have “largely been improved in my opinion.” quoting the first Reviewer, “All comments have been addressed” as the second Reviewer states.

Concerning the detailed comments, we address them below and list the corresponding changes in the manuscript. Original comments of the first Referee are in blue and changes in the manuscript are in red, both here and in the revised manuscript.

With these changes and clarification, we trust our manuscript is now suitable for publication in PLOS One.

Reply to the first Referee:

Referee 1: The diagonal of the scatter plot matrix could be used to show the distribution of the two classes. It would add valuable

information to the figure. An example of such a plot can be found at, for example, https : //seaborn.pydata.org/examples/scatterplotmat

Reply: We agree with the Referee and we thank him/her for the reference he/she provided us with. We added the below scatter plot matrix.

Action: The below figure was added along with the sentence “To visualize the outliers, the six features were reduced to three using the Principle Component Analysis (PCA). The correlations between each new dimension and the two others were then illustrated using a two-dimensional scatter plot matrix shown in Fig. 1. The diagonal plots show the univariate distribution of each dimension.”

Correlation plot of the buildings samples after applying PCA for dimension reduction, with outliers highlighted in blue.

Referee 1: It would be, additionally, be very informative to have a map similar to figures 3 and 4 with the buildings that are actually used in the analysis.

Reply: Indeed we agree that the suggested map would inform the reader about the used buildings in the analysis.

Action: The below figure was added along with the sentence “To visualize the outliers, the six features were reduced to three using the Principle Component Analysis (PCA). The correlations between each new dimension and the two others were then

illustrated using a two-dimensional scatter plot matrix shown in Fig. 1. The diagonal plots show the univariate distribution of each dimension. The spatial distribution of the buildings used in the development of the predictive algorithms is shown in Fig. 2.”

0 0.5 1 2 Km

Excluded Buildings

Buildings Used

2

 Spatial distribution of the accepted buildings after the data pre-processing.

Referee 1: Since all information is contained in figure 2, I would recommend removing figure 1, as it does not provide any useful additional information. But this I leave to the authors.

Reply/Action:

We agree with the Referee that the information showed in the two figures is redundant, especially after updating the scatter plot. The figure illustrating the 3D correlations between PCA dimensions was removed.

Referee 1: Please provide a few references to the added part in lines 136-137 to justify the percentages used.

Reply: We thank the Referee for highlighting the need for references. The paper of A. Clark entitled “The machine learning

audit- CRISP-DM Framework” was added.

Action: The reference was added along with the sentence “The dataset of 1, 536 samples was subdivided into training, valida- tion, and test sets each containing respectively 859, 369, and 308 samples, which correspond respectively to the 55%, 25%, and 20% splits, often recommended in the literature [31].”

Referee 1: In line 202 you mention that you recover known properties about heights. Please add a sentence what these properties are or add references.

Reply/Action: We thank the Referee for asking for clarification. Indeed the sentence is not clear.

Action: We added the following to the text: The fact that the distribution of predicted buildings’ heights follows a power-law and not a log-normal is a confirmation that our model recovers known properties about the heights; namely that they follow a

power-law and not a log-normal distribution.

Referee 1: I apologize reiterating this point again, but in the case of buildings, what does it mean that the underlying processes are either multiplicative or additive? I personally have no intuition what that means in terms of building heights. Please clarify further.

Reply: We thank the Referee for his/her care for clarity. In terms of distribution, we a power-laws mathematically arise when the underlying process is multiplicative. This reference includes the mathematical details of our statement. Mitzenmacher, Michael. “A brief history of generative models for power law and lognormal distributions.” Internet mathematics 1.2 (2004): 226-251.

Action: Rereading the statement we made about the multiplicative processes we see that it has no relevance to the flow of idea and thus decided to remove it.

Referee 1: I appreciate that you added the section from line 100 and onward. However, it does still not clarify why MLF-NNs are a good choice for the analysis you did. I’m no expert in machine learning, so please make this point a little clearer for people like me.

Reply: Many machine learning algorithms are available with different architectures. In our manuscript, we chose three well- known algorithms (linear regression, NN, and RF). Each model with a given architecture learns its parameters based on the training set. After that, to evaluate the model’s performance, and thus to choose the best among them, a performance metric is applied to compare how close the actual data is to the model’s prediction. This metric is normally a measure of error, or how far the predictions are from the actual data and thus the algorithm with the best metric value is chosen. After running the algorithms with our dataset, the NN algorithm described in the manuscript performed better than all the other. Therefore, it was considered the best choice for our analysis.

Action: The following sentence was added at the beginning of the Methods section: Many machine learning algorithms are available with different architectures. In our manuscript, we chose three well-known algorithms (linear and logistic regression, NN, and RF) that are described in more details below. Each model with a given architecture learns its parameters based on the training set. After that, to evaluate the model’s performance, and thus to choose the best among them, a performance metric is applied to compare how close the actual data is to the model’s prediction. This metric is normally a measure of error, or how far the predictions are from the actual data and thus the algorithm with the best metric value is chosen.

3

---

## [Decision Letter · Decision Letter 2]

14 Jan 2021

Machine learning for buildings' characterization and

power-law recovery of urban metrics

PONE-D-20-10968R2

Dear Dr. Najem,

We’re pleased to inform you that your manuscript has been judged scientifically suitable for publication and will be formally accepted for publication once it meets all outstanding technical requirements.

Kind regards,

Celine Rozenblat

Academic Editor

PLOS ONE

Additional Editor Comments (optional):

Reviewers' comments:

Reviewer's Responses to Questions

**Comments to the Author**

1. If the authors have adequately addressed your comments raised in a previous round of review and you feel that this manuscript is now acceptable for publication, you may indicate that here to bypass the “Comments to the Author” section, enter your conflict of interest statement in the “Confidential to Editor” section, and submit your "Accept" recommendation.

Reviewer #1: All comments have been addressed

Reviewer #2: All comments have been addressed

2. Is the manuscript technically sound, and do the data support the conclusions?

Reviewer #1: Yes

Reviewer #2: Yes

3. Has the statistical analysis been performed appropriately and rigorously? 

Reviewer #1: Yes

Reviewer #2: Yes

4. Have the authors made all data underlying the findings in their manuscript fully available?

Reviewer #1: No

Reviewer #2: (No Response)

5. Is the manuscript presented in an intelligible fashion and written in standard English?

Reviewer #1: Yes

Reviewer #2: Yes

6. Review Comments to the Author

Reviewer #1: All my questions have been clearly addressed. I thank the authors for their careful revisions and clarifications!

Reviewer #2: The authors already in the previous iteration addressed my comments. Now, under consideration of the comments from the other reviewer, the manuscript further improved.

7. PLOS authors have the option to publish the peer review history of their article (what does this mean?). If published, this will include your full peer review and any attached files.

Reviewer #1: No

Reviewer #2: No

---

## [Editor Report · Acceptance letter]

18 Jan 2021

PONE-D-20-10968R2 

Machine learning for buildings’ characterization and power-law recovery of urban metrics 

Dear Dr. Najem:

I'm pleased to inform you that your manuscript has been deemed suitable for publication in PLOS ONE. Congratulations! Your manuscript is now with our production department. 

Kind regards, 

on behalf of

Prof. Celine Rozenblat 

Academic Editor

PLOS ONE